# Schema-learning and rebinding as mechanisms of in-context learning and emergence

**Sivaramakrishnan Swaminathan**   **Antoine Dedieu**   **Rajkumar Vasudeva Raju**
**Murray Shanahan**   **Miguel Lázaro-Gredilla**   **Dileep George**
Google DeepMind
{sivark,adedieu,rajvraju,mshanahan,lazarogredilla,dileepgeorge}@google.com

## Abstract

In-context learning (ICL) is one of the most powerful and most unexpected capabilities to emerge in recent transformer-based large language models (LLMs). Yet the mechanisms that underlie it are poorly understood. In this paper, we demonstrate that comparable ICL capabilities can be acquired by an alternative sequence prediction learning method, namely clone-structured causal graphs (CSCGs). A key property of CSCGs is that, unlike transformer-based LLMs, they are *interpretable*, which considerably simplifies the task of explaining how ICL works. We show that ICL in CSCG uses a combination of (a) learning template (schema) circuits for pattern completion, (b) retrieving relevant templates in a context-sensitive manner, and (c) rebinding novel tokens to appropriate slots in the templates. We go on to marshall evidence for the hypothesis that similar mechanisms underlie ICL in LLMs. For example, we find that, with CSCGs as with LLMs, different capabilities emerge at different levels of overparameterization, suggesting that overparameterization helps in learning more complex template (schema) circuits. By showing how ICL can be achieved with small models and datasets, we open up a path to novel architectures, and take a vital step towards a more general understanding of the mechanics behind this important capability.

## 1   Introduction

In a pre-trained sequence model, *in-context learning* (ICL), or *few-shot prompting*, is the ability to learn a new task from a small set of examples presented within the context (the prompt) at inference time. Surprisingly, large language models (LLMs) trained on sufficient data exhibit ICL, even though they are trained only with the objective of next token prediction [1, 2]. A good deal of the ongoing excitement surrounding LLMs arises from this unexpected capacity, since it dramatically enlarges their set of potential applications. Ongoing attempts to understand this capability take a variety of forms, including higher-level normative accounts using Bayesian inference [3], and mechanistic explanations involving implicit gradient descent [4] or induction heads [5]. Despite this, the mechanisms that underlie ICL in LLMs remain somewhat mysterious.

We take an alternative approach, studying a sequence-learning model called a clone-structured causal graph (CSCG) [6, 7] to reveal the conditions that drive ICL. We show that ICL can be explained as a combination of (a) learning template circuits for pattern completion, (b) retrieving relevant templates in a context-sensitive manner, and (c) rebinding of novel tokens to appropriate slots in templates [8]. Unlike n-gram models, CSCGs allow transitive generalization in the latent space: they assign semantically sensible non-zero probabilities to sequences never seen during training to ensure that the contexts (prompts) used for retrieval are not pure memorizations. In addition, the binding of novel tokens to slots in learned templates allows the same structural knowledge to be applied to entirely novel inputs. We hypothesize how similar mechanisms could exist in transformer-based LLMs. By

37th Conference on Neural Information Processing Systems (NeurIPS 2023).

elucidating the principles that underpin the mechanics of ICL, we hope to pave the way for the design of novel architectures for abstraction and generalization, while the building blocks we identify guide the search for mechanistically interpretable [9] and editable [10] circuits in transformers [11].

## 2 Rebinding algorithm for clone-structured causal graphs

### 2.1 Background on clone-structured causal graphs (CSCGs)

Consider an agent executing a series of discrete actions $a_1, \ldots, a_{N-1}$ with $a_n \in \{1, \ldots, N_{\text{actions}}\}$, e.g. walking in a room. As a result of each action, the agent receives a perceptually aliased observation [12], resulting in the stream of random variables $X_1, \ldots, X_N$ with observed values $x_1, \ldots, x_N$, where each $x_n \in \{1, \ldots, N_{\text{obs}}\}$. CSCG [6] is a probabilistic sequence learning model that introduces a latent explanatory variable $Z_n$ at each timestep $n$, with values $z_n \in \{1, \ldots, N_{\text{latent}}\}$, to model the action-conditional stream of observations as

$$P(x_1, \ldots, x_N | a_1, \ldots, a_{N-1}) = \sum_{z_1, \ldots, z_n} P(x_1 | z_1) P(z_1) \prod_{n=2}^{N} P(x_n | z_n) P(z_n | z_{n-1}, a_{n-1}).$$

A transition tensor $T : T_{ijk} = P(Z_n = k | Z_{n-1} = j, a_{n-1} = i) \, \forall n$ represents the action-conditional dynamics. $T$ defines a directed multigraph, whose nodes correspond to the values of $z$. Conditioned on an action, each entry of $T$ is the weight of a directed edge between two nodes (from the row index to the column index of that entry). A CSCG can thus recover a graph that represents the latent causal structure [13] of the environment (see Fig. 1D for an example) which can then be used for planning.

An emission matrix $E : E_{ij} = P(X_n = j | Z_n = i) \, \forall n$ represents the observation probabilities. CSCGs have a deterministic observation model: for any latent value $z$, the same observation $x$ is always emitted. Multiple values of $z$ can result in the same observed $x$, making the model overcomplete [14]. The restriction to deterministic $E$ makes CSCGs less general than a hidden Markov model (HMM), but easier to learn [6]. A CSCG can disambiguate multiple aliased percepts (same observation $x$) into distinct causes (different latent values $z$) given a sufficiently long context. If the observations correspond to word tokens, CSCGs can also be used as a language model, with a single action that accesses the next token[1].

### 2.2 Rebinding in CSCGs

On encountering a new environment with a similar structure but different observations, an agent can learn that environment faster by reusing the latent graph $T$ from prior experience and relearning just the emission matrix, through a *rebinding* process [15]. Rebinding can be interpreted as a soft intervention on the agent's model [16, 17]. See Fig. 1D & F for examples of two rooms that share the same latent structure but different observations. When a new emission matrix binds to an existing schema, it has to respect the *clone structure* of the original emission matrix (Fig. 1E). The clone structure function $\mathcal{C}(\cdot) \in 1, \ldots, N_{\text{obs}}$ partitions the latent state in $N_{\text{obs}}$ *slots*: two latent states $z = i$ and $z = i'$ belong to the same slot iff $\mathcal{C}(i) = \mathcal{C}(i')$. An emission matrix respects the clone structure $\mathcal{C}$ if $\mathcal{C}(i) = \mathcal{C}(i') \implies E_{ij} = E_{i'j} \, \forall i, i', j$. The 3-tuple $\{T, \mathcal{C}, E\}$ defines a *grounded schema*, the tuple $\{T, \mathcal{C}\}$ defines an *ungrounded schema with clone structure* and $T$ alone is a *schema* [15].

#### 2.2.1 Fast rebinding by attending to surprise

Often, environment changes are localized such that most of the latent structure and observation mapping is preserved while just a few observations need to be rebound: for example, just replacing the carpet in a room while the wall colors remain the same, or getting exposed to a new word in a familiar context. This insight can be utilized to derive an algorithm that focuses the update of the emission matrix only to those observations that were found surprising by the existing model.

Suppose that at test time, a grounded schema $\{T, \mathcal{C}, E^0\}$ is exposed to a sequence with novel observations. Algorithm 1 proposes a fast procedure to update the emission matrix to the new observations by only performing local updates, and to bind the updated emission matrix to the existing schema $T$, defining a new grounded schema $\{T, \mathcal{C}, E^{\text{rb}}\}$. We call this process *fast rebinding*.

---

[1]This can be generalized to include actions that skip over one or more next tokens.

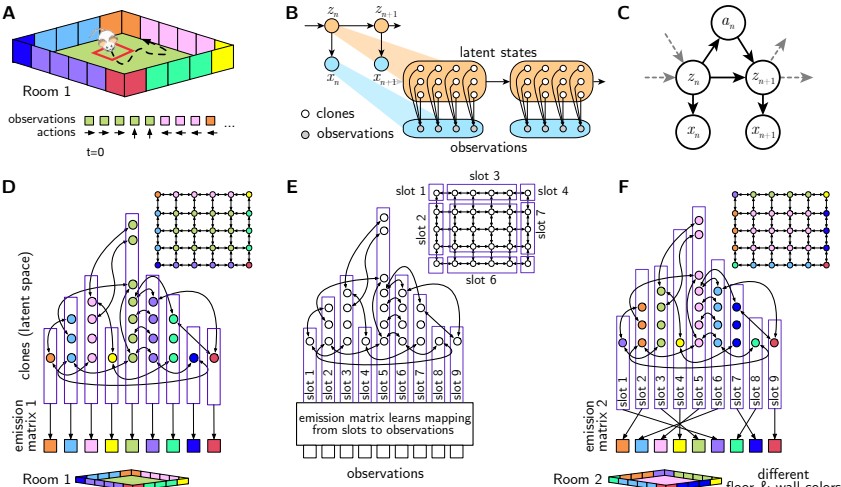

Figure 1: **A**. Inducing room structure (*cognitive maps*) from sequential sensory observations is challenging due to perceptual aliasing – local observations do not identify locations uniquely. **B**. Cloned hidden Markov models (HMMs) [7]. Each observation is mapped to multiple clone states in the latent space. **C**. Graphical model for CSCGs [6], extending cloned HMMs by incorporating actions. CSCGs utilize the latent space to overcome the perceptual aliasing problem. Different clones learn to represent different temporal contexts to recover the latent structure of the room. **D**. Learned CSCG for the room shown in panel A consists of a latent transition matrix and an emission matrix. We visualize the model in two ways: (i) stacking clone states for respective observations into columns, and (ii) clones as nodes in a transition graph, colored with their respective emissions. **E**. The emission matrix imposes a *slot* structure – nodes within the same slot are constrained to bind to the same observation. A new environment with the same latent structure but different observation mapping (Room 2) can be learned quickly by freezing the transition graph and slot structure, and learning a new emission matrix by rebinding slots to a new set of observations. **F**. CSCG for a *different room* learned purely through rebinding.

Given a prompt $(x_1, \ldots, x_N)$ and a surprise threshold, Algorithm 1 proceeds by (a) identifying emission matrix entries that need updating then (b) updating these entries using the Expectation-Maximization (EM) algorithm [18]. The conditional probability $P(X_n = j \mid x_{\setminus n})$ of tokens at timestep $n$ given all other tokens is used to identify timesteps and latent states that are surprising. Step 3 identifies *anchors*, i.e., latent states corresponding to observations that are correctly predicted with high confidence: anchors are not rebound. Step 4 identifies *candidates for rebinding* as latent states (a) not among the anchor states and (b) corresponding to timesteps at which observations are incorrectly predicted with high confidence. Finally, instead of re-learning the whole emission matrix [15, Appendix A.2], Step 5 (detailed in Appendix A) *locally updates* the emission matrix by only applying EM on the latent states and timesteps identified in Step 4. As a result, only a small subset of rows differ between $E^0$ and $E^{\mathrm{rb}}$. Protected rows correspond to either (a) anchors in the current prompt or (b) slots not relevant to the current prompt but possibly relevant to future observations. Section 6 discusses how a similar mechanism could be implemented in transformers.

---

**Algorithm 1** – Fast rebinding algorithm

---

**Input:** Grounded schema $\{T, \mathcal{C}, E^0\}$, pseudocount $\epsilon$, prompt $(x_1, \ldots, x_N)$, surprise probability threshold $p_{\mathrm{surprise}}$.

**Output:** Rebound emission matrix $E^{\mathrm{rb}}$

1: Define $\tilde{E}^0 \propto E^0 + \epsilon$, with normalized rows.

2: For timestep $n$, use the emission matrix $\tilde{E}^0$ to compute
$P(X_n = j \mid x_{\setminus n}) = P(X_n = j \mid x_1, \ldots, x_{n-1}, x_{n+1}, \ldots, x_N), \ \forall j \leq N_{\mathrm{obs}}$

3: Identify latent states and timesteps that can act as anchors:
$\mathcal{A} = \left\{(i, n) \ \middle| \ P(X_n = x_n \mid x_{\setminus n}) > p_{\mathrm{surprise}}, \ \text{and} \ \mathcal{C}(i) = x_n\right\}$

4: Identify latent states to be rebound (and their timesteps):
$\mathcal{R} = \left\{(i, n) \ \middle| \ P(X_n = j \mid x_{\setminus n}) > p_{\mathrm{surprise}}, \ j \neq x_n, \ (\cdot, n) \notin \mathcal{A}, \ (i, \cdot) \notin \mathcal{A} \ \text{and} \ \mathcal{C}(i) = j\right\}$

5: Fix $T$, and use EM to update the emission matrix (initialized with $E^0$, and without using any pseudocount) by only using the beliefs for latent states $i$ and timesteps $n$ such that $(i, n) \in \mathcal{R}$.

---

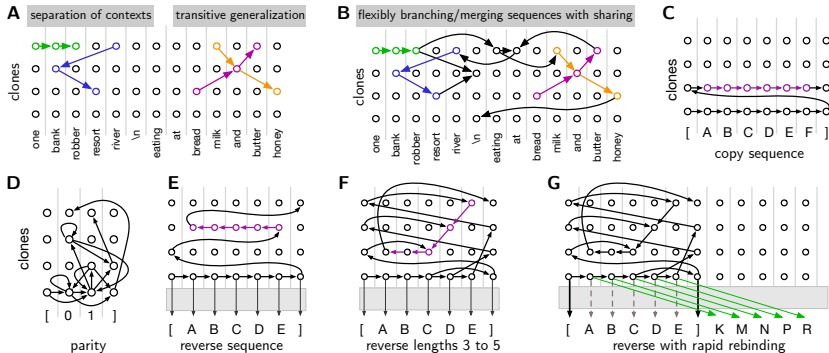

Figure 2: **A**. CSCGs allow both separation of contexts and transitive generalization. The word "bank" is wired to multiple clones corresponding to the different contexts it is used in. If "milk and honey", and "bread and butter" are seen in training, transitive generalization occurs if they get wired through the same "and" clone: "bread and honey" and "milk and butter" appear as valid sequences. **B**. Probabilistic branching & merging of sequences. **C** – **F**. Exemplar CSCG circuits for copying a sequence, parity operation, reversing a list with exactly five elements, reversing lists with a variable number of elements. **G**. Rebinding to new observations: dashed gray arrows correspond to old emissions while green arrows correspond to newly rebound emissions.

After rebinding, we complete the prompt by performing MAP inference conditioned on the provided prompt in the rebound CSCG. We run the max-product algorithm [19] forward (the backward messages are all uniform) thus generating a series of MAP observations for the tokens following the prompt. We stop once we generate a delimiter token. See Algorithm 2 in Appendix B for details.

## 3    Outline of the overall argument using CSCG

### 3.1    Context-dependent latent representations and transitive generalization

The clone structure of CSCGs allows context-based separation and appropriate blending for language modeling. For example, the sense of the word "bank" in "bank robber" is different from the one in "river bank". CSCG learning disambiguates these contexts in the latent space by wiring them to different clones to improve predictive accuracy. In Fig. 2A, the sentences "river bank resort", and "one bank robber" use different clones of "bank". Sequences can have probabilistic branching: "one bank robber" can terminate at "\n", or continue to "eating at river bank resort" or "eating bread and honey", or "eating bread and butter at river bank resort" (Fig. 2B). CSCGs also allow the merging of contexts that result in transitive generalization: even if training data has only the sequences "bread and butter", and "milk and honey", if they go through the same clone state "and", the model will generalize to "bread and honey" and "milk and butter", assigning non-zero probability to those sequences. Due to the combination of context-sensitive separation and transitivity, related topics, concepts, and algorithms get clustered into sub-networks that pass through the same clones. A prompt's context would activate its sub-network, and transitive generalization allows for prompts that are not exact memorizations. As we show in Section 4, the Bayesian inference perspective on ICL [3] corresponds to this context-sensitive and transitively generalizing storage and retrieval alone, and is insufficient to explain the ICL properties we consider in the next sections.

### 3.2    Learning flexible schemas (template circuits) and rebinding

Just like learning room layouts, CSCG can learn automata circuits [20] for sequence-to-sequence (seq2seq) algorithms. See Fig. 2 for CSCG circuits for computing parity, copying a sequence, and reversing sequences of multiple lengths. The list reversal circuit in Fig. 2E is bound to the specific symbols $A, B, C, D, E$ used in training. For use as a template, slots in this graph must be able to appropriately bind to contents (arbitrary symbols) that occur in context at test time [8, 21]. The rebinding mechanism (formalized in Algorithm 1) can intuitively be understood as operating based on prediction errors – when the latent context strongly predicts the latent state corresponding to a time instant, but the actual observation is mismatched, rebinding adjusts the emission matrix to wire all the clones of that latent state to the surprising observation. Such a mechanism to mix and gate previous knowledge with new content allows circuits learned during training to become flexible templates

with slots that can dynamically bind to new inputs as required. For example, in the list reversal schema in Fig. 2F, tokens "[", and "]" are prior contents that detect the beginning and end of the list – these act as anchors for grounding the schema in the observations. Probabilistic branching based on the end of list token "]" allows for length generalization, whereas absorbing arbitrary symbols into the slots corresponding to $A, B, C, D, E$ allows the algorithm to generalize to new symbols. Fig. 2G illustrates the outcome of this rebinding mechanism where the slots emitting $A, B, C, D, E$ are respectively rebound to symbols $K, M, N, P, R$ from the input prompt. Similarly, in the sentence "I wrote in a notebook using a dax", rebinding can absorb the new token "dax" into the context by binding it to a clone corresponding to "pencil" or "pen", and use the new word in those contexts.

### 3.3 Instruction-based or content-based retrieval and completion of tasks

**Zero-shot task recognition as content-based retrieval using rebinding:** Many striking examples of zero-shot learning involve recognizing tasks from prompts, and repeating them on new inputs. For example, given a prompt "Input: [p, q, r, s] Output: [p, p, q, q, r, r, s, s]; Input: [l, m, n, o] Output: [l, l, m, m, n, n, o, o]" LLMs can infer the task as repeating the elements of the sequence, and apply that to complete the output for a new input prompt even when the tokens "p, q, r, s, l, m, n, o" were not seen during training, in association with this task. The rebinding mechanism offers a natural explanation for this. Given the prompt, expectation maximization (EM) [18] simultaneously evaluates the different rebindings to multiple latent algorithm schemas to infer the best binding, which is then applied to complete the query prompt.

**Instruction-based retrieval:** When algorithms are trained with prefixed language instructions, CSCGs learn instruction sub-networks that directly point to the circuits that represent the algorithms (see Section 4.2). The algorithm can be retrieved by direct prompting with language instructions that can be significantly different from training instructions due to transitive generalization and rebinding.

### 3.4 Emergence

We hypothesize and empirically demonstrate in Section 4, that emergence is explainable as the combined effects of the above properties (context-separation, transitive generalization, schema-formation, and rebinding), model capacity, and patterns in the data. Training on a bigger dataset results in the induction of more templates that might not have occurred in the smaller dataset. Learning the schematic circuits for more complex algorithms or more patterns in the data requires greater model capacity because overparameterization helps in the optimization process.

## 4 Results

We substantiate the above argument using empirical results on three datasets: (a) the GINC benchmark introduced in [3], (b) a suite of algorithm learning tasks that we introduce in our LIALT datasets, and (c) a zero-shot word usage induction task on a CSCG language model.

### 4.1 Context-sensitive retrieval on GINC dataset matches Bayesian inference explanation

**Dataset:** The GINC dataset [3] introduced for studying ICL, is generated from a uniform mixture of five factorial HMMs [22]. Each factorial HMM is referred to as a *concept*. A document is created by concatenating independent sentence samples from a concept. The in-context test prompts have examples of lengths $k \in \{3, 5, 8, 10\}$, varying in number from $n = 0$ to $n = 64$, with 2500 prompts for each setting $(k, n)$. Each prompt uniformly selects a concept, samples $n - 1$ examples $x_{:k}^{(1)}, \ldots, x_{:k}^{(n-1)}$ of length $k$, and one example $x_{:k-1}^{(n)}$ of length $k - 1$. The in-context task is to infer the most likely last token of the last example, i.e., $\mathrm{argmax}_{x_{k-1}^{(n)}} \, p\left(x_{k-1}^{(n)} | x_{:k}^{(1)}, \ldots, x_{:k}^{(n-1)}, x_{:k-1}^{(n)}\right)$. Since the vocabulary is shared among different latent concepts, observations in GINC are aliased like in natural language, and solving the task requires the model to disambiguate the aliased observations to correctly infer the latent concepts.

**Training:** We train a single CSCG with 50 clones on the GINC dataset for 100 full-batch EM iterations using a pseudocount [6] of $\epsilon = 10^{-2}$. Given a test prompt, CSCG infers the most likely hidden sequence for that prompt, then predicts the next most likely observation.

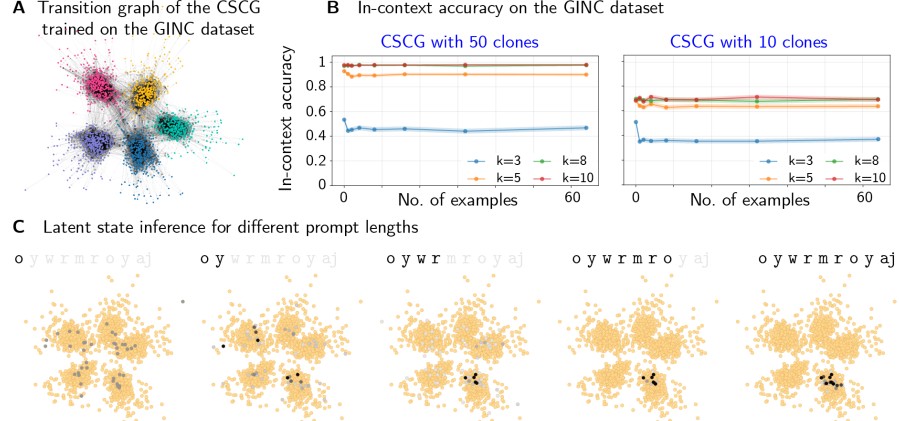

Figure 3: **A**. Visualizing the transition graph of a CSCG with 50 clones trained on the GINC dataset from [3]. The clones cluster into five groups – one per *concept*. **B**.[Left] In-context accuracy averaged over the GINC test dataset (with 95% confidence intervals (CIs) as in [3]), for the same model For contexts of 8 and 10 tokens, the model predicts the most likely next token at least 95% of the time—including in the zero-shot regime. [Right] In-context accuracy decreases when we reduce the number of clones to 10—for $k \in \{8, 10\}$ it drops from above 95% to below 75%. The numerical values are reported in Appendix C, Table 1. **C**. Decoded latent state distributions (increasing intensities of black for higher density) for the CSCG with 50 clones, for an $n = 0$ & $k = 10$ prompt "o y w r m r o y aj", when truncated to different lengths ($k = 2, 3, 5, 8, 10$). Longer prompts improve latent state estimation—resulting in better concept retrieval, and next token prediction.

**Results:** CSCG learns different latent sub-networks corresponding to the five latent concepts in the GINC dataset ( Fig. 3A), and inference on a supplied prompt retrieves the correct latent sub-network (Fig. 3C). Increasing the prompt length improves the localization of the sub-network and the particular states within the sub-network. Figure 3C visualizes the decoded latent state distribution for an example prompt in the zero-shot setting ($n = 0$). The decoding starts out uncertain, and improves as the prompt gets longer. This localization (on the graph) results in effective schema retrieval, and hence accurate prompt completion. Figure 3B[left] reports the in-context accuracy—defined as the average ratio of correct predictions—for each $(k, n)$ pair of the GINC test set. CSCG in-context accuracy matches the patterns exhibited by LSTMs and transformers in [3], while slightly improving their performance. Fig. 3B also shows that a CSCG with larger capacity, i.e. with 50 clones per token, better separates the latent concepts and significantly outperforms a CSCG with only 10 clones per token. Fig. 9[left] in Appendix C displays the CSCG in-context confidence: for larger contexts, CSCG is better at disambiguating aliasing and the averaged predictions probabilities are higher. Finally, Fig. 9[right] shows that similarly to the transformer and LSTM in [3], CSCG fails at ICL when test prompts are sampled from concepts unseen during training. The GINC results match the context-based retrieval argument in Section 3.1: ICL in this setting is the retrieval of a shared latent concept between the prompt and the model. By using the long-range coherence of concepts in the training documents, the model learns to separate concepts into different latent representations. Despite the train and prompt distribution mismatch [3], CSCG succeeds at prompt completion because the representation allows transitive mixing.

## 4.2 Learning schemas for seq2seq algorithms and generalization using rebinding

**Training dataset:** To test the ability of CSCG to learn *algorithms* that generalize to novel inputs not seen during training, we construct the Language Instructed Algorithm Learning Tasks (LIALT) dataset. The LIALT training set contains demonstrations of 13 list and matrix algorithms displayed in Fig. 4A[top-left]. A demonstration consists of a multi-word language instruction—each algorithm has five different instructions—followed by 10 input-output examples of that algorithm. See Tables 2 & 3 in Appendix D.1 for the complete list of instructions used. For each instruction, the dataset contains 20 demonstrations. Within a demonstration, the language instruction and the examples are separated by a "/" delimiter. Demonstrations are separated by a "\n" delimiter. The input lists and matrices values are created by uniformly sampling from a vocabulary of 676 tokens, created by random pairings of uppercase letters. List operations examples vary in length from 3 to 6, and the matrix operations are of sizes $2 \times 2$ or $3 \times 3$. Fig. 4A [bottom-left] shows the training data format.

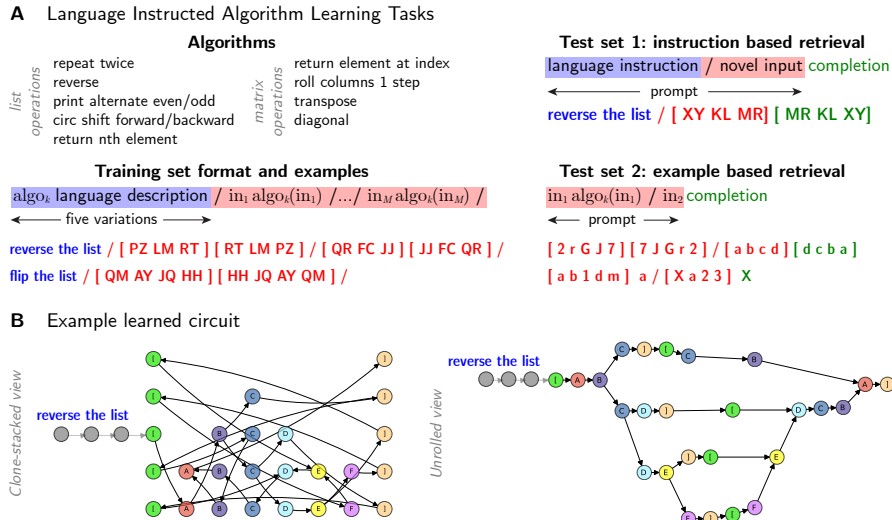

Figure 4: **A**. [Top-left] List and matrix algorithms used in the LIALT dataset. Format of the training set [bottom-left] and examples of the two LIALT test sets [right]. **B**. Example of a learned circuit for the "reverse" algorithm, displayed by stacking clones [left] or unrolling them [right].

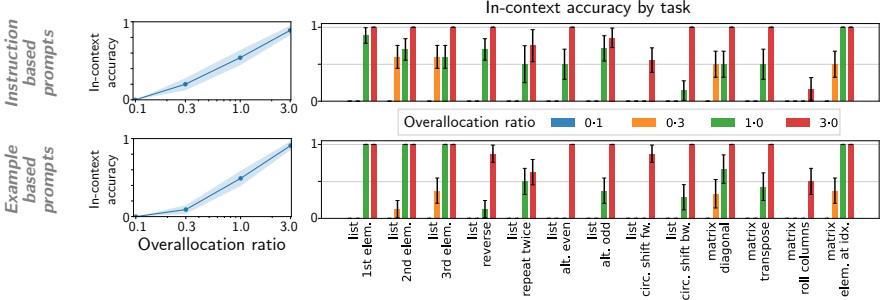

Figure 5: [Left] In-context accuracy (ICA) with $95\%$ CIs after a single EM iteration, as a function of the overallocation ratio for a CSCG trained on LIALT and averaged [top] on the instruction-based LIALT test set [bottom] on the example-based LIALT test set. ICA increases with model capacity. [Right] ICA with standard errors per task on the two LIALT test sets: for each task, overparametrization improves performance. Invisible bars indicate zero accuracy for the respective combinations of model and task. All the numerical values are in Appendix D.3. Figure 11 in the Appendix visualizes the same quantities after EM convergence; the similarity demonstrates that the fast rebinding algorithm is not just localized in its updates, but also rapid.

**Test dataset:** LIALT has two test datasets, respectively containing: (a) instruction-based retrieval prompts, and (b) example-based retrieval prompts. An instruction-based retrieval test prompt consists of a natural language instruction followed by a single input. An example-based retrieval test prompt consists of a first input-output example of an algorithm, without any natural instruction, followed by a second input. All the lists and matrices in the two test datasets contain novel tokens. For both types of prompts, the in-context task is to predict the algorithm's output when applied to the (last) input. Note that for an example-based prompt, CSCG has to infer the algorithm used from the first example. Each test set contains 100 prompts, constructed by uniformly sampling instructions, and list or matrix tokens. Fig. 4A [right] shows the formats of these two test sets.

**Training:** For each token, a CSCG allocates its number of clones proportionally to the number of distinct contexts in the training data in which it occurs[2]. We parameterize CSCG capacity via this proportionality factor – the "overallocation ratio". We train CSCGs for an increasing sequence of overallocation ratios on the training data with $500$ EM iterations and a pseudocount of $\epsilon = 10^{-6}$. After running EM, we run 10 iterations of Viterbi training [23].

---

[2]As the same token might occur in different contexts in the training data, knowing the context allows predicting the sequence of following tokens, up to the next "/" delimiter.

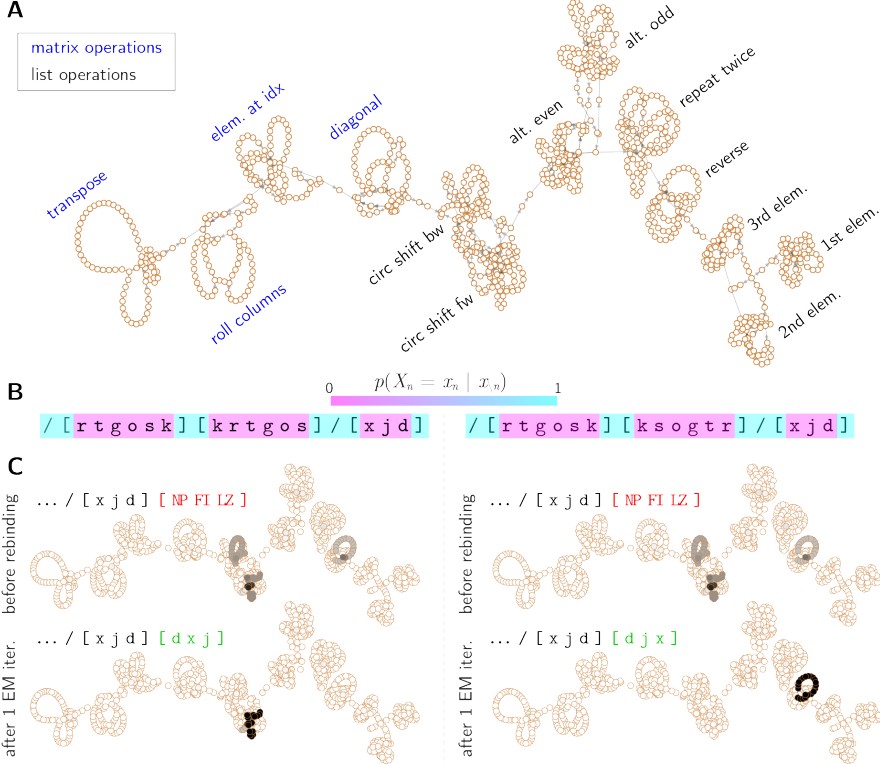

Figure 6: **A.** Transition graph of the CSCG model learned on the LIALT dataset, visualized using the Kamada-Kawai algorithm. **B.** Visualizing the inferred probability of the observation at timestep $n$, conditioned on observations at all other timesteps, before rebinding. This drives the identification of anchors and slots selected for rebinding. **C.** Decoded latent state distributions (and predicted prompt completions) for the two different example-based LIALT prompts specified in subfig. B: (top) before rebinding, and (bottom) after one iteration of EM. Fig. 12 in Appendix D.3.2 extends the same visualization to EM convergence. The left prompt corresponds to the operation of circularly shifting the list forward, and the right prompt corresponds to reversing the list.

**Results:** CSCGs with sufficient model capacity successfully learn the algorithms in the training set, and rebinding generalizes those algorithms to novel tokens. Fig. 4B shows the extracted circuit for the list reversal algorithm. Fig. 5[left] presents the in-context accuracy of CSCGs (using $\epsilon = 10^{-6}$ and $p_{\text{surprise}} = 0.1$) on the two LIALT test sets: the best performing CSCG (a) successfully rebinds the learned schemas to the test prompts' novel tokens and (b) correctly infers the algorithm from a single input-output pair for example-based prompts. Fig. 5 also shows that model size drives ICL performance [left] even when breaking down the performance by tasks [right].

The learned CSCG (initialized with an overallocation ratio of 3) is visualized in Fig. 10 in the Appendix, using stacked clones. Fig. 6A shows the transition graph using the Kamada-Kawai algorithm [24]. It reveals thirteen loosely connected clusters corresponding to the thirteen algorithms present in the LIALT dataset. Fig. 6B illustrates the rebinding process, with the decoded distributions over latent states of the learned CSCG model, for two different example-based prompts. Even before any rebinding, the identification of anchors and slots already restricts the decoding to schemas compatible with the prompt *structure*—in this case based on brackets & delimiters. However, the structure is insufficient to disambiguate completely between the compatible schemas (list operations corresponding to reversal, circular forward shift, and circular backward shift), and both the chosen prompts result in the same latent state distribution. Hence, the decoded distribution after the first E-step localizes to the three compatible schemas. In the M-step that follows, the slots in all three schemas will be rebound for this prompt. At the end of the first EM iteration, the new bindings for slots in the correct schema will be highly certain given the consistent evidence, while inconsistent evidence will lead to uncertain bindings for the other slots. In the E-step of the second iteration, the respective levels of certainty in the bindings then help promote the correct algorithm schema to become the most likely decoding—and complete the prompt appropriately. Note that a single EM step is sufficient to derive the correct rebinding in these examples. Compare Figs. 5 & 11, and the

tables in Appendix Sec. D.3 for how the in-context completion performance after the first EM step in the rebinding process is very similar to that at the end of EM convergence.

The LIALT results substantiate the arguments we made in Sections 3.2 and 3.3. Bayesian inference of the latent context based on long-term coherence (sufficient for the GINC results in Section 4.1) does not explain the remapping of a latent representation to completely new tokens as required for generalizing on the LIALT algorithms. Without rebinding, even a prompt containing a full-length example of an algorithm but with novel tokens does not retrieve the correct algorithm schema or produce the correct completion based on inference over the latent states alone (Fig. 6B, first row). By contrast, simultaneously inferring the rebindings and the latent states results in accurate retrieval of the algorithm schema and the correct prompt completion (Fig. 6B, second row). CSCGs are thus able to learn seq2seq algorithms and generalize those algorithms to novel tokens using rebinding.

**Emergence:** ICL performance of CSCG on the LIALT dataset shows characteristics attributed to emergence. In-context accuracy has a clear dependency on the level of overparameterization of CSCG, offering evidence in support of our hypothesis in Section 3.4.

### 4.3 Dax test

In language, the "dax" test [25] is used to demonstrate the capability of a model to absorb the usage of an entirely new word from a single presentation. To test for this capability, we train a CSCG on the PreCo dataset [26], which is a large-scale English dataset for coreference resolution. We then test the model on five word-replaced query prompts, where certain words in the prompts do not appear in the training set. We use Algorithm 1 with $\epsilon = 10^{-6}$ and $p_{\text{surprise}} = \frac{1}{16}$ to rebind the emission matrix on each of these prompts, each time probing the model for completing a sentence by filling in the blanks (uncertain inputs) using MAP inference. Fig. 7 shows these results.

| unseen word | | Replacement query prompts | after rebinding | Fill in the blanks probing |
|---|---|---|---|---|
| warming | -> heating | ... make global heating worse ! | | global heating may be a big problem, ... |
| winners | -> victors | ... victors receive a new computer ... | | contest victors will be announced ... |
| planets | -> terras | ... have n't found life on other terras yet . | | seven other terras also go around the sun . |
| artificial | -> AG | ... the field of AG intelligence . | | .... dangerous in this AG intelligence progress then ? |
| bikes | -> cycles | ... most people ride cycles to school . | | ... more people ride their cycles around the world . |

Figure 7: Examples of the dax test on a CSCG trained on the PreCo dataset. In each row, the novel word in red (e.g. "terras") is absorbed by binding it to the clones of the corresponding word in blue (e.g. "planets"). The CSCG can then use the new token in similar contexts, as demonstrated by the fill-in-the-blanks probing.

## 5 Related work

**In-context learning:** Similar to how humans learn by analogy [27] and how synaptic plasticity allows the brain to rapidly adapt to a new task [28], ICL capabilities [1] allows a pre-trained model to learn a new task given only a few examples. [29, 30] showed how demonstrations that explicitly guide the reasoning process improve the ICL performance of transformers on new complex tasks. We clarify below some concepts that should not be confused with ICL, and then discuss some works that aim at understanding ICL and the factors that influence it.

**Supervised learning (SL) and few-shot learning (FSL):** SL approaches learn a mapping that minimizes a loss on the training data: gradient methods are a popular paradigm [31, 32, 33]. In FSL, a model learns to rapidly adapt to a new task from a limited number of supervised examples [34, 35, 36], and performs this same task at inference. In contrast, ICL tasks are only revealed at inference. [37, 38] showed that finetuning transformers on ICL instructions improves their ICL performance.

**Meta-learning:** The meta-learning paradigm aims at learning to adapt to a new task with only a few examples [39, 40, 41] by using multiple learning experiences. In contrast, ICL directly emerges from the pre-trained model. [42, 43] proposed a meta-learning framework for ICL where the model is fine-tuned: it learns to leverage few-shot examples and to adapt to new tasks at inference time.

**How ICL works:** [3] explained ICL as implicit Bayesian inference and constructed the GINC dataset (see Section 4.1) for demonstrating ICL. [44] abstracted ICL as an algorithm learning problem and found that a transformer can implicitly infer a hypothesis function. Similarly, [45] showed that a transformer can be trained to perform ICL of unseen linear functions, with performance

comparable to the optimal least squares estimator. [46] showed that, in the linear case, transformers implicitly implement gradient descent and train an implicit linear model on the ICL examples. [4] proposed a dual between transformer attention and gradient methods and suggested pre-trained models as meta-optimizers. They presented ICL as implicit finetuning, where the forward pass on the demonstrative examples produces meta-gradients. Finally, [5] showed the existence of "induction heads" in transformers, that emerge during training, copy previous patterns, and drive ICL capacities.

**What influences ICL:** [1, 47] indicated that LLMs' ICL performance "emerges" then keeps improving when the model size increases. [48] proposed a substitute for the positional encoding, and demonstrated how transformers can learn schemas for algorithmic tasks and generalize to test sequences longer than any seen during training. Some works have highlighted the role of the training data in ICL. [49] showed that ICL emerge when the training data has a large number of rare classes and when examples appear in clusters. while [50] demonstrated that ICL emerge when a model is trained on a combination of multiple corpora, and that low perplexity and ICL performance do not always correlate. [51, 52] found that ICL is highly unstable and is influenced by the prompting template, the selection of in-context examples, and the order of the examples. [53] showed that the ICL performance is driven by the exposure to the label space, the input distribution, and the overall format of the sequence. Similarly, [54] found that selecting ICL examples with closer embeddings to ICL test sample improves ICL performance, and [55] showed that adding explanations in-context improves performance. Finally, [56] recently claimed that the sharp emergence of ICL in larger models might be an artifact of the metrics, not a fundamental property of the model.

## 6 Discussion

With ICL deconstructed into schema learning, schema retrieval, and slot rebinding, an interesting avenue for future work would be to probe various sequence models for how robustly each of these components are manifested – or even construct models around these principles. Here we consider how this framework might map to transformers, where the phenomenon of ICL was originally observed. Unlike CSCGs, transformers buffer the inputs and represent location as positional encoding, allowing attention to gate by the structure of the prompt, along with the contents. Prior explanations [3, 4] do not distinguish the role of sequence positions vis-a-vis contents; we argue that theories might need to emphasize this distinction (see Fig. 8A) to fully understand the inductive biases behind ICL. We conjecture (see Fig. 8B) that layers of the transformer implement multiple mixed templates of positions and content, evaluated at different offsets of a prompt. The template assembly that can auto-regressively match the prompt wins out the competition to gate the content. The rebinding mechanism requires only a few iterations of sparse updates to the emission matrix, and can be temporally "unrolled" into a forward pass, allowing ICL behavior with fixed weights since the slotting process lives in the space of activations.

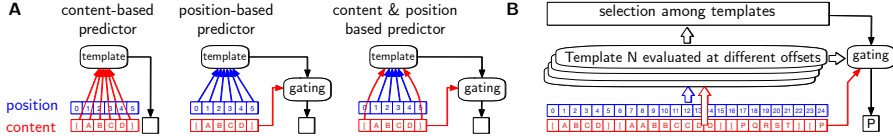

Figure 8: **A**. Learned templates in a transformer could involve content, position, or a mix of both. **B**. Activations in the forward pass of a transformer could be selected among pre-learned templates that mix content and position to achieve ICL without weight changes.

Coming back to CSCGs, implementations can scale to larger models and datasets by exploiting sparsity and parallelizing computations in the EM steps. Allowing a factorized latent space and adding skip connections would also allow compositionality while enabling scalability. Further, while we have illustrated here the concept of rebinding to attach new symbols to existing slots, rebinding "through time" can also target connections between clones, enabling compositional behavior in-context. We leave these explorations for future research. Our goal here has been to elucidate a general framework for ICL behavior, leveraging the interpretability of CSCGs. We hope this demystifies the ICL behavior observed in LLMs by analogy, showcases avenues for further research on ICL capabilities, and provides broad impetus for interpretable methods.

## Acknowledgements

We thank Stephanie Chan, Andrew Lampinen, Anirudh Goyal, Dharshan Kumaran, Neel Nanda and Guangyao Zhou for helpful discussions and comments on the draft.

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

# A  Locally updating the emission matrix with the transition matrix fixed

We reuse the same notations as in [15], Appendix A.2. The authors describe the EM algorithm for learning the emission matrix of a CSCG with a fixed transition matrix. In particular their M step defines the new emission matrix as:

$$E(j) = \sum_{n=1}^{N} 1_{X_n=j} \, \gamma(n) \oslash \sum_{n=1}^{N} \gamma(n), \; \forall j$$

where $E(j)$ is a column of the emission matrix corresponding to the emission $j$, $1_{X_n=j}$ is an indicator function, $\oslash$ is the element-wise division and $\gamma(n)$ is derived by the authors from the forward and backward probabilities. The $i$ entry of the vector $E(j)$ is then defined as:

$$E_{ij} = \frac{\sum_{n=1}^{N} 1_{X_n=j} \, \gamma_i(n)}{\sum_{n=1}^{N} \gamma_i(n)}.$$

In contrast, in Section 2.2, Step 5 of Algorithm 1 only updates the row $i$ for which we can find a pair $(i, n) \in \mathcal{R}$, by only using the beliefs at timestep $n$. For this row, the $j$th entry becomes:

$$E_{ij} = \frac{\sum_{n: \, (i,n)\in\mathcal{R}} 1_{X_n=j} \, \gamma_i(n)}{\sum_{n: \, (i,n)\in\mathcal{R}} \gamma_i(n)}.$$

The pseudocount used in Step 1 of Algorithm 1 is an uncertainty parameter that lets the model smooth over incorrect observations. More details of this parameter are available in [6].

# B  Prompt completion algorithm

Algorithm 2 describes the prompt completion algorithm introduced in Section 2.2. It implicitly considers a single action, which takes the next sequence element.

---

**Algorithm 2** – Prompt completion

**Input:** Grounded schema $\{T, \mathcal{C}, E^{\mathrm{rb}}\}$ with rebound CSCG emission matrix $E^{\mathrm{rb}}$, delimiter token $x_\emptyset$, prompt $x^{\mathrm{(prompt)}} = (x_1, \ldots, x_N)$

**Output:** A completed prompt $x^{\mathrm{(completed)}} = (x_1, \ldots, x_N, x_{N+1}, \ldots, x_{N+P} = x_\emptyset)$

1: Run max-product for MAP inference and return $z^{\mathrm{MAP}} = (z_1, \ldots, z_N) = \mathrm{argmax}_z \, P(z|x^{\mathrm{(prompt)}})$.

2: Set $\ell = 0$. While $x_{N+\ell} \neq x_\emptyset$, increment $\ell \leftarrow \ell + 1$ and sample the next most likely observation: $z_{N+\ell} \in \mathrm{argmax}_j \, T_{z_{N+\ell-1}, \, j}$ and $x_{N+\ell} \in \mathrm{argmax}_j \, E^{\mathrm{rb}}_{z_{N+\ell}, \, j}$.

---

## C   Additional materials for the GINC dataset

First, we present two additional plots for the GINC experiment.

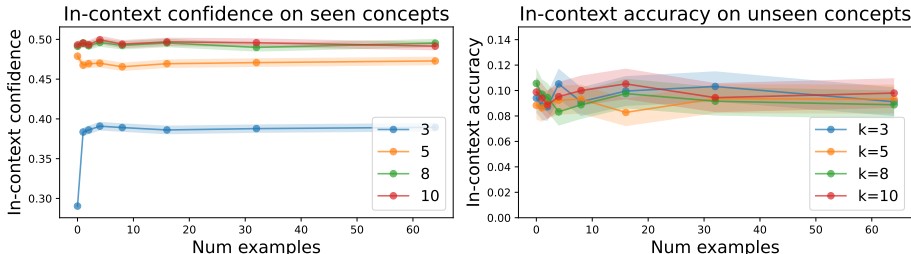

Figure 9: [Left] In-context confidence for the CSCG with 50 clones on the GINC test dataset, defined as the averaged probability of the predictions. For larger values of $k$, CSCG correctly infers the context of the aliased observations and is more confident in its predictions. [Right] Similar to the transformer and LSTM reported in [3], CSCG fails to extrapolate and has a low in-context accuracy when the test prompts are sampled from five novel concepts, unseen during training.

Second, we present the table of results associated with Fig. 3 for the CSCGs with 10 and 50 clones.

| Context length | No. of examples | CSCG with 10 clones | CSCG with 50 clones |
|---|---|---|---|
| 3 | 0 | 0.509 (0.020) | 0.534 (0.020) |
|   | 1 | 0.351 (0.019) | 0.445 (0.019) |
|   | 2 | 0.366 (0.019) | 0.453 (0.020) |
|   | 4 | 0.356 (0.019) | 0.468 (0.020) |
|   | 8 | 0.360 (0.019) | 0.454 (0.020) |
|   | 16 | 0.354 (0.019) | 0.460 (0.020) |
|   | 32 | 0.354 (0.019) | 0.441 (0.0219) |
|   | 64 | 0.369 (0.019) | 0.468 (0.020) |
| 5 | 0 | 0.682 (0.018) | 0.927 (0.010) |
|   | 1 | 0.640 (0.019) | 0.927 (0.012) |
|   | 2 | 0.629 (0.019) | 0.904 (0.012) |
|   | 4 | 0.654 (0.019) | 0.883 (0.013) |
|   | 8 | 0.627 (0.019) | 0.894 (0.012) |
|   | 16 | 0.637 (0.019) | 0.902 (0.012) |
|   | 32 | 0.634 (0.019) | 0.901 (0.012) |
|   | 64 | 0.637 (0.019) | 0.899 (0.012) |
| 8 | 0 | 0.696 (0.018) | 0.969 (0.007) |
|   | 1 | 0.694 (0.018) | 0.972 (0.007) |
|   | 2 | 0.686 (0.018) | 0.972 (0.006) |
|   | 4 | 0.681 (0.018) | 0.978 (0.006) |
|   | 8 | 0.690 (0.018) | 0.973 (0.006) |
|   | 16 | 0.686 (0.018) | 0.975 (0.006) |
|   | 32 | 0.676 (0.018) | 0.968 (0.006) |
|   | 64 | 0.694 (0.018) | 0.975 (0.007) |
| 10 | 0 | 0.684 (0.018) | 0.975 (0.006) |
|   | 1 | 0.705 (0.018) | 0.977 (0.006) |
|   | 2 | 0.674 (0.018) | 0.971 (0.006) |
|   | 4 | 0.713 (0.018) | 0.974 (0.006) |
|   | 8 | 0.690 (0.018) | 0.977 (0.006) |
|   | 16 | 0.689 (0.018) | 0.977 (0.006) |
|   | 32 | 0.712 (0.018) | 0.978 (0.006) |
|   | 64 | 0.690 (0.018) | 0.978 (0.006) |

Table 1: In-context accuracy for a CSCG with 10 clones and a CSCG 50 clones trained on the GINC dataset, averaged (with $95\%$ confidence intervals) on each each pair $(k, n)$ of context length and number of examples $n$ of the GINC test set.

**CSCG performs better on zero-shot prompts than on few-shot prompts:** We observe that, for short contexts, CSCG in-context accuracy is higher on zero-shot prompts $n = 0$ than on few-shot prompts $n = 1, 2, \ldots$. We hypothesize that the difference between the training and the prompt distributions creates a gap that lowers few-shot in-context accuracy. The performance gap disappears

for larger contexts $k \in \{8, 10\}$ as they "overpower" the train-test distribution divergence. [3] made a similar observation for transformers. However, their performance gap was also observable for larger contexts.

# D  Additional materials for the LIALT dataset

## D.1  Natural language instructions

Tables 2 and 3 present the natural language instructions respectively used for the nine list algorithms and four matrix algorithms of the LIALT dataset. Language instructions are grouped in clusters of five: all five instructions within one cluster describe to the same algorithm. As described in the main text, each demonstration of the LIALT training and first test set uniformly selects one instruction.

```
"find the element at index zero of the list"        "print the element at index one of the list"
"print the first element from the list"             "find the second element from the list"
"return the leading element from the list"          "retrieve the second element from the list"
"find the head element from the list"               "locate the second item from the list"
'retrieve the starting element from the list'       "return the element in second place from the list"
```
```
"print the element at index two of the list"        "reverse the list"
"find the third element from the list"              "mirror the list"
"locate the third element from the list"            "flip the list"
"output the third item from the list"               "flip the order of the list"
"return the element in third place from the list"   "reverse the order of the items in the list"
```
```
"duplicate each list item"                          "rotate the list elements one place forward"
"replicate every element in the list"               "roll the list elements one position to the right"
"make a copy of each element in the list"           "switch the items of the list one position forward"
"clone each element in the list"                    "advance the list elements one index forward"
"create a second instance of every element in the   "move the list elements one position forward"
list"
```
```
"print every other member in the list starting with "print every other member in the list starting with
the second member"                                  the first member"
"retrieve alternate items in the list starting with "find alternate elements in the list beginning with
the second item"                                    the first element"
"return every other object in the list starting with "print every second item in the list, starting with
the second object"                                  the first element"
"retrieve every other entry in the list starting with "output every second element in the list, starting
the second entry"                                   from the first element"
"output odd indexed elements"                       "output even indexed elements"
```
```
"rotate the list elements one place backward"
"move the list elements one position to the left"
"change the items of the list one position backward"
"displace the elements of the list one index backward"
"roll the list items one position backward"
```

Table 2: Natural language instructions for the list algorithms used in the LIALT dataset

```
"return the matrix diagonal"                         "return the matrix transpose"
"collect the diagonal values of the matrix"          "retrieve the transpose of the matrix"
"retrieve the diagonal elements of the matrix"       "get the transposed matrix"
"return the diagonal entries of the matrix"          "compute the transposed form of the matrix"
"fetch the diagonal items of the matrix"             "derive the transpose matrix"
```
```
"roll the columns of the matrix to the right"        "find the matrix element in the second row and second
                                                     column"
"rotate the matrix columns to the right"             'find the value in the second row and second column of
                                                     the matrix"
"move the matrix columns to the right"               "fetch the matrix element located in row 2 and column
                                                     2"
"shift the columns of the matrix to the right"       "print the value at 2 2 in the matrix"
"spin the matrix columns to the right"               "retrieve the matrix element at 2 2"
```

Table 3: Natural language instructions for the matrix algorithms used in the LIALT dataset

## D.2  Learned CSCG model

Our next Figure 10 displays the transition graph of the CSCG model trained on the LIALT dataset with an overallocation ratio of 3, with stacked clones for each symbol.

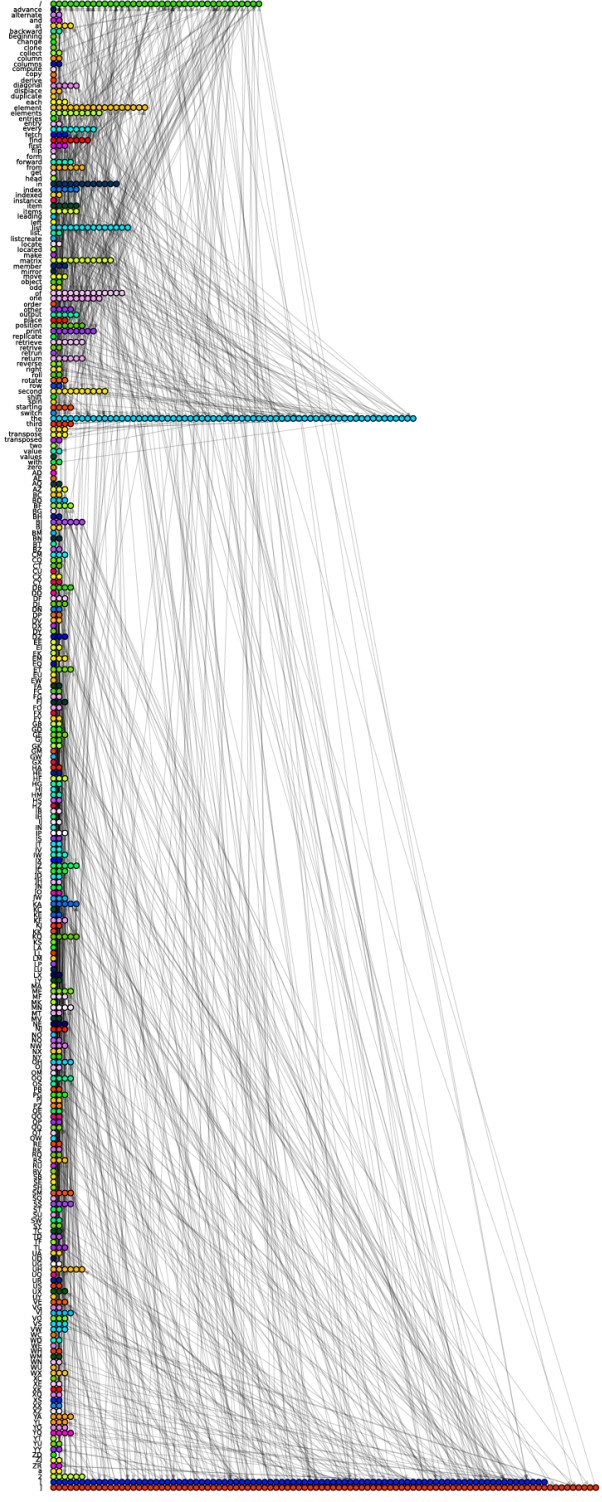

Figure 10: CSCG model learned on the LIALT dataset, visualized with stacked clones.

### D.3 Results on the LIALT dataset

#### D.3.1 After a single EM iteration

Presented below are the tables of results associated with Fig. 5. Table 4 contains the in-context accuracies averaged on the entire test set, Table 5 contains the in-context accuracies per task on instructions-based prompts, and Table 6 contains the in-context accuracies per task on example-based prompts.

| Overallocation ratio | Instruction-based prompts | Example-based prompts |
|---|---|---|
| 0.1 | 0.00 (0.00) | 0.00 (0.00) |
| 0.3 | 0.20 (0.08) | 0.09 (0.06) |
| 1.0 | 0.54 (0.10) | 0.49 (0.10) |
| 3.0 | 0.89 (0.06) | 0.91 (0.06) |

Table 4: Average in-context accuracy of each CSCG model—with 95% confidence intervals—as a function of CSCG overallocation on both (a) the instruction-based LILAT test set and (b) the example-based LIALT test set.

| Task | Overallocation ratio | | | |
|---|---|---|---|---|
| | 0.1 | 0.3 | 1.0 | 3.0 |
| list 1st elem. | 0.00 (0.00) | 0.00 (0.00) | 0.89 (0.10) | 1.00 (0.00) |
| list 2nd elem. | 0.00 (0.00) | 0.60 (0.15) | 0.70 (0.14) | 1.00 (0.00) |
| list 3rd elem. | 0.00 (0.00) | 0.60 (0.15) | 0.60 (0.15) | 1.00 (0.00) |
| list reverse | 0.00 (0.00) | 0.00 (0.00) | 0.70 (0.14) | 1.00 (0.00) |
| list repeat twice | 0.00 (0.00) | 0.00 (0.00) | 0.50 (0.25) | 0.75 (0.22) |
| list alt. even | 0.00 (0.00) | 0.00 (0.00) | 0.50 (0.20) | 1.00 (0.00) |
| list alt. odd | 0.00 (0.00) | 0.00 (0.00) | 0.71 (0.17) | 0.86 (0.13) |
| list circ. shift fw. | 0.00 (0.00) | 0.00 (0.00) | 0.00 (0.00) | 0.56 (0.17) |
| list circ. shift bw. | 0.00 (0.00) | 0.00 (0.00) | 0.14 (0.13) | 1.00 (0.00) |
| matrix diagonal | 0.00 (0.00) | 0.50 (0.18) | 0.50 (0.18) | 1.00 (0.00) |
| matrix transpose | 0.00 (0.00) | 0.00 (0.00) | 0.50 (0.20) | 1.00 (0.00) |
| matrix roll columns | 0.00 (0.00) | 0.00 (0.00) | 0.00 (0.00) | 0.17 (0.15) |
| matrix elem. at idx. | 0.00 (0.00) | 0.50 (0.18) | 1.00 (0.00) | 1.00 (0.00) |

Table 5: Average in-context accuracy by task—with standard errors—as a function of CSCG overallocation on instruction-based prompts.

| Task | Overallocation ratio | | | |
|---|---|---|---|---|
| | 0.1 | 0.3 | 1.0 | 3.0 |
| list 1st elem. | 0.00 (0.00) | 0.00 (0.00) | 1.00 (0.00) | 1.00 (0.00) |
| list 2nd elem. | 0.00 (0.00) | 0.12 (0.12) | 1.00 (0.00) | 1.00 (0.00) |
| list 3rd elem. | 0.00 (0.00) | 0.38 (0.17) | 1.00 (0.00) | 1.00 (0.00) |
| list reverse | 0.00 (0.00) | 0.00 (0.00) | 0.12 (0.12) | 0.88 (0.12) |
| list repeat twice | 0.00 (0.00) | 0.00 (0.00) | 0.50 (0.18) | 0.62 (0.17) |
| list alt. even | 0.00 (0.00) | 0.00 (0.00) | 0.00 (0.00) | 1.00 (0.00) |
| list alt. odd | 0.00 (0.00) | 0.00 (0.00) | 0.38 (0.17) | 1.00 (0.00) |
| list circ. shift fw. | 0.00 (0.00) | 0.00 (0.00) | 0.00 (0.00) | 0.88 (0.12) |
| list circ. shift bw. | 0.00 (0.00) | 0.00 (0.00) | 0.29 (0.17) | 1.00 (0.00) |
| matrix diagonal | 0.00 (0.00) | 0.33 (0.19) | 0.67 (0.19) | 1.00 (0.00) |
| matrix transpose | 0.00 (0.00) | 0.00 (0.00) | 0.43 (0.19) | 1.00 (0.00) |
| matrix roll columns | 0.00 (0.00) | 0.00 (0.00) | 0.00 (0.00) | 0.50 (0.18) |
| matrix elem. at idx. | 0.00 (0.00) | 0.38 (0.17) | 1.00 (0.00) | 1.00 (0.00) |

Table 6: Average in-context accuracy by task—with standard errors—as a function of CSCG overallocation on example-based prompts.

#### D.3.2 After EM convergence

Figs. 11 & 12 present analogues of Figs. 5 & 6 but after the EM algorithm in the rebinding process has converged. We note that the results are mostly identical. Table 7 contains the in-context accuracies averaged on the entire test set, Table 8 contains the in-context accuracies per task on instructions-based prompts, and Table 9 contains the in-context accuracies per task on example-based prompts.

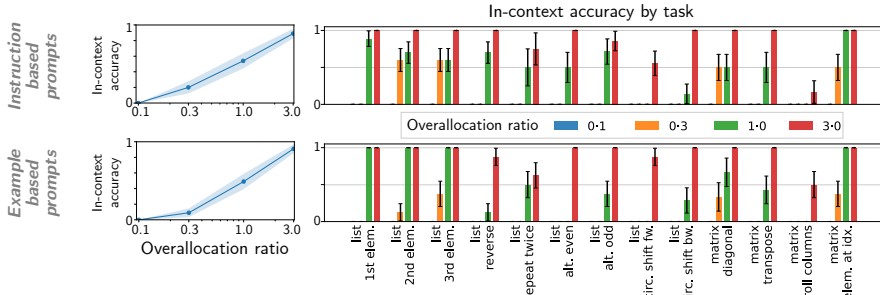

Figure 11: [Left] In-context accuracy (with 95% CIs) after EM convergence, as a function of the overallocation ratio for a CSCG trained on LIALT and averaged [top] on the instruction-based LIALT test set [bottom] on the example-based LIALT test set. In-context accuracy increases for CSCGs with larger capacities. [Right] In-context accuracy (with standard errors) per task on the two LIALT test sets: for each task, overparametrization improves performance. All the numerical values are in Appendix D.3. Invisible bars indicate zero accuracy for the respective combination of model and task.

| Overallocation ratio | Instruction-based prompts | Example-based prompts |
|---|---|---|
| 0.1 | 0.00 (0.00) | 0.00 (0.00) |
| 0.3 | 0.16 (0.07) | 0.11 (0.06) |
| 1.0 | 0.54 (0.10) | 0.49 (0.10) |
| 3.0 | 0.89 (0.06) | 0.93 (0.05) |

Table 7: Average in-context accuracy of each CSCG model—with 95% confidence intervals—as a function of CSCG overallocation on both (a) the instruction-based LILAT test set and (b) the example-based LIALT test set.

| | Overallocation ratio | | | |
|---|---|---|---|---|
| Task | 0.1 | 0.3 | 1.0 | 3.0 |
| list 1st elem. | 0.00 (0.00) | 0.00 (0.00) | 0.89 (0.10) | 1.00 (0.00) |
| list 2nd elem. | 0.00 (0.00) | 0.60 (0.15) | 0.70 (0.14) | 1.00 (0.00) |
| list 3rd elem. | 0.00 (0.00) | 0.60 (0.15) | 0.60 (0.15) | 1.00 (0.00) |
| list reverse | 0.00 (0.00) | 0.00 (0.00) | 0.70 (0.14) | 1.00 (0.00) |
| list repeat twice | 0.00 (0.00) | 0.00 (0.00) | 0.50 (0.25) | 0.75 (0.22) |
| list alt. even | 0.00 (0.00) | 0.00 (0.00) | 0.50 (0.20) | 1.00 (0.00) |
| list alt. odd | 0.00 (0.00) | 0.00 (0.00) | 0.71 (0.17) | 0.86 (0.13) |
| list circ. shift fw. | 0.00 (0.00) | 0.00 (0.00) | 0.00 (0.00) | 0.56 (0.17) |
| list circ. shift bw. | 0.00 (0.00) | 0.00 (0.00) | 0.14 (0.13) | 1.00 (0.00) |
| matrix diagonal | 0.00 (0.00) | 0.00 (0.00) | 0.50 (0.18) | 1.00 (0.00) |
| matrix transpose | 0.00 (0.00) | 0.00 (0.00) | 0.50 (0.20) | 1.00 (0.00) |
| matrix roll columns | 0.00 (0.00) | 0.00 (0.00) | 0.00 (0.00) | 0.17 (0.15) |
| matrix elem. at idx. | 0.00 (0.00) | 0.50 (0.18) | 1.00 (0.00) | 1.00 (0.00) |

Table 8: Average in-context accuracy by task—with standard errors—as a function of CSCG overallocation on instruction-based prompts.

| | Overallocation ratio | | | |
|---|---|---|---|---|
| Task | 0.1 | 0.3 | 1.0 | 3.0 |
| list 1st elem. | 0.00 (0.00) | 0.00 (0.00) | 1.00 (0.00) | 1.00 (0.00) |
| list 2nd elem. | 0.00 (0.00) | 0.12 (0.12) | 1.00 (0.00) | 1.00 (0.00) |
| list 3rd elem. | 0.00 (0.00) | 0.38 (0.17) | 1.00 (0.00) | 1.00 (0.00) |
| list reverse | 0.00 (0.00) | 0.00 (0.00) | 0.00 (0.00) | 0.88 (0.12) |
| list repeat twice | 0.00 (0.00) | 0.00 (0.00) | 0.50 (0.18) | 0.88 (0.12) |
| list alt. even | 0.00 (0.00) | 0.00 (0.00) | 0.00 (0.00) | 1.00 (0.00) |
| list alt. odd | 0.00 (0.00) | 0.00 (0.00) | 0.50 (0.18) | 1.00 (0.00) |
| list circ. shift fw. | 0.00 (0.00) | 0.00 (0.00) | 0.00 (0.00) | 0.88 (0.12) |
| list circ. shift bw. | 0.00 (0.00) | 0.00 (0.00) | 0.29 (0.17) | 1.00 (0.00) |
| matrix diagonal | 0.00 (0.00) | 0.67 (0.19) | 0.67 (0.19) | 1.00 (0.00) |
| matrix transpose | 0.00 (0.00) | 0.00 (0.00) | 0.43 (0.19) | 1.00 (0.00) |
| matrix roll columns | 0.00 (0.00) | 0.00 (0.00) | 0.00 (0.00) | 0.50 (0.18) |
| matrix elem. at idx. | 0.00 (0.00) | 0.38 (0.17) | 1.00 (0.00) | 1.00 (0.00) |

Table 9: Average in-context accuracy by task—with standard errors—as a function of CSCG overallocation on example-based prompts.

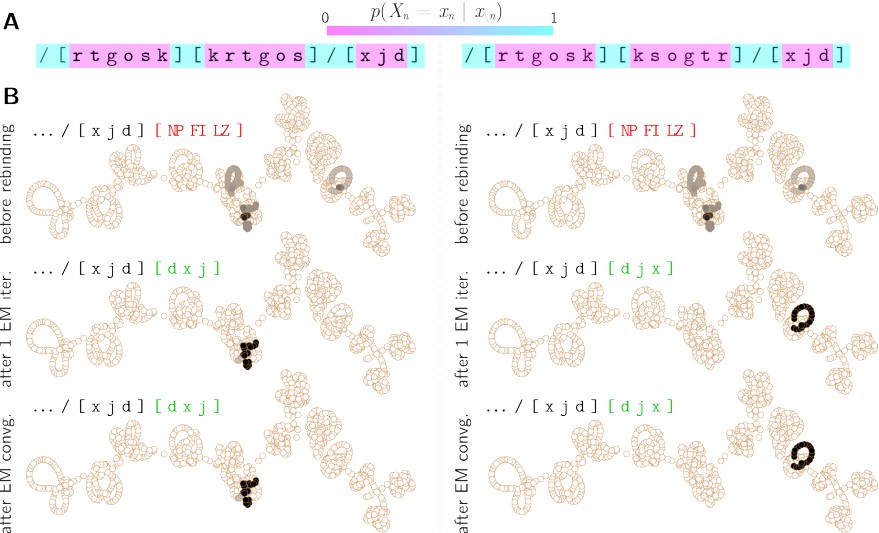

Figure 12: Extension of Fig. 6 to EM convergence. **A.** Visualizing the inferred probability of the observation at timestep $n$, conditioned on observations at all other timesteps, before rebinding. This drives the identification of anchors and slots selected for rebinding. **B.** Decoded latent state distributions — and predicted prompt completions — for the two different example-based LIALT prompts specified in subfig. A: (top) before rebinding, (middle) after one iteration of EM, and (bottom) after EM convergence. The left prompt corresponds to the operation of circularly shifting the list forward, and the right prompt corresponds to reversing the list.

## D.4 Example failures

Finally, we present a few examples which illustrate the failure modes of our approach. These are primarily a consequence of imperfections in the learned CSCG model. Each example is presented in the format (prompt, ground truth correct output, actual model response).

1. For these failures, the instruction circuit has been wired to the wrong algorithm circuit (possibly driven by the ambiguity of the forward slash delimiter separating the instruction from the example), resulting in the retrieval of the wrong schema.

   - output odd indexed elements / [ U V B Q K I ]
     [ U B K ] /
     [ V Q I ] /
   - flip the list / [ S E J ]
     [ J E S ] /
     [ S S E E J J ]
   - reverse the list / [ R T B ]
     [ B T R ] /
     [ R R T T B B ] /
   - mirror the list / [ B A O T ]
     [ T O A B ] /
     [ B B A A O O T T ] /

2. For these failures, the schema has been learned incorrectly.

   - switch the items of the list one position forward / [ L N G X M T ]
     [ T L N G X M ] /
     [ T L N G X M T ] [ T L N G X M T ] ...
   - shift the columns of the matrix to the right / [ [ D Y ] [ V F ] ]
     [ [ Y D ] [ F V ] ] /
     [ [ get
   - / [ Z J B ] [ Z Z J J B B ] / [ B A E F W L ]
     [ B B A A E E F F W W L L ] /
     [ B B A E F F W W L L ] /
   - / [ V P X T ] [ P T ] / [ V F J P E W ]
     [ F P W ] /
     [ F P W ] [ F P W ] ...