## A  Prompt completion algorithm

Algorithm 2 describes the prompt completion algorithm introduced in Section 2.2. It implicitly considers a single action, which takes the next sequence element.

---
**Algorithm 2** – Prompt completion

---
**Input:** Grounded schema $\{T, \mathcal{C}, E^{\mathrm{rb}}\}$ with rebound CSCG emission matrix $E^{\mathrm{rb}}$, delimiter token $x_\emptyset$, prompt $x^{(\mathrm{prompt})} = (x_1, \ldots, x_m)$

**Output:** A completed prompt $x^{(\mathrm{prompt\ completed})} = (x_1, \ldots, x_m, x_{m+1}, \ldots, x_{m+p} = x_\emptyset)$

1: Run max-product for MAP inference and return $z^{\mathrm{MAP}} = (z_1, \ldots, z_m) = \operatorname{argmax}_z p(z|x^{(\mathrm{prompt})})$.

2: Set $\ell = 0$. While $x_{m+\ell} \neq x_\emptyset$, increment $\ell \leftarrow \ell + 1$ and sample the next most likely observation: $z_{m+\ell} \in \operatorname{argmax}_j T_{z_{m+\ell-1},\, j}$ and $x_{m+\ell} \in \operatorname{argmax}_j E^{\mathrm{rb}}_{z_{m+\ell},\, j}$.

---

## B  Rapid binding in CSCGs

Algorithm 3 is a variant of the rebinding Algorithm 1 that does not use EM. Instead, it first searches for "surprising observations": a surprise has a low probability of being emitted by its decoded clone. This decoded clone (and all the clones in its clone set) are then *rapidly bound* to emit the surprise.

---
**Algorithm 3** – Rapid binding algorithm

---
**Input:** Grounded schema $\{T, \mathcal{C}, E^0\}$, pseudocount $\epsilon$, surprise probability $p_{\mathrm{surprise}}$, prompt $x^{(\mathrm{prompt})}$

**Output:** Rapidly bound emission matrix $E^{\mathrm{rb}}$

1: Add the pseudocount $\epsilon$ to the initial emission matrix and normalize its rows.

2: Run max-product for MAP inference and return $z^{\mathrm{MAP}} = \operatorname{argmax}_z p(z|x^{(\mathrm{prompt})})$.

3: Define the set of surprising observations $\mathcal{S} = \left\{ x_n : E^0_{z^{\mathrm{MAP}}_n,\, x_n} \leq p_{\mathrm{surprise}} \right\}$.

4: Set $E^{\mathrm{rb}} = E^0$. For each surprising observation $x_n$, (rapidly) bind $z^{\mathrm{MAP}}_n$ and all the clones in the clone slot of $z^{\mathrm{MAP}}_n$ to emit $x_n$.
That is, $\forall \tilde{z} : \mathcal{C}(\tilde{z}) = \mathcal{C}(z^{\mathrm{MAP}}_n)$ set $E^{\mathrm{rb}}_{\tilde{z},\, x_n} = 1$ and $E^{\mathrm{rb}}_{\tilde{z},\, j} = 0$, $\forall j \neq x_n$.

---

Initially, for any given clone $i$, only one entry $j$ of the row $E^0_i$ in the emission matrix is set to 1. After Step 1, we have $E^0_{i,\, j} = \frac{1}{1+N_{\mathrm{obs}}\epsilon}$ and $E^0_{i,\, k} = \frac{\epsilon}{1+N_{\mathrm{obs}}\epsilon}$, $\forall k \neq j$. We can then use $p_{\mathrm{surprise}} = \frac{1}{2(1+N_{\mathrm{obs}}\epsilon)}$.

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

 trained on the LIALT dataset, displayed by both (a) stacking clones (b) unrolling them using the Kamada-Kawai algorithm [20].

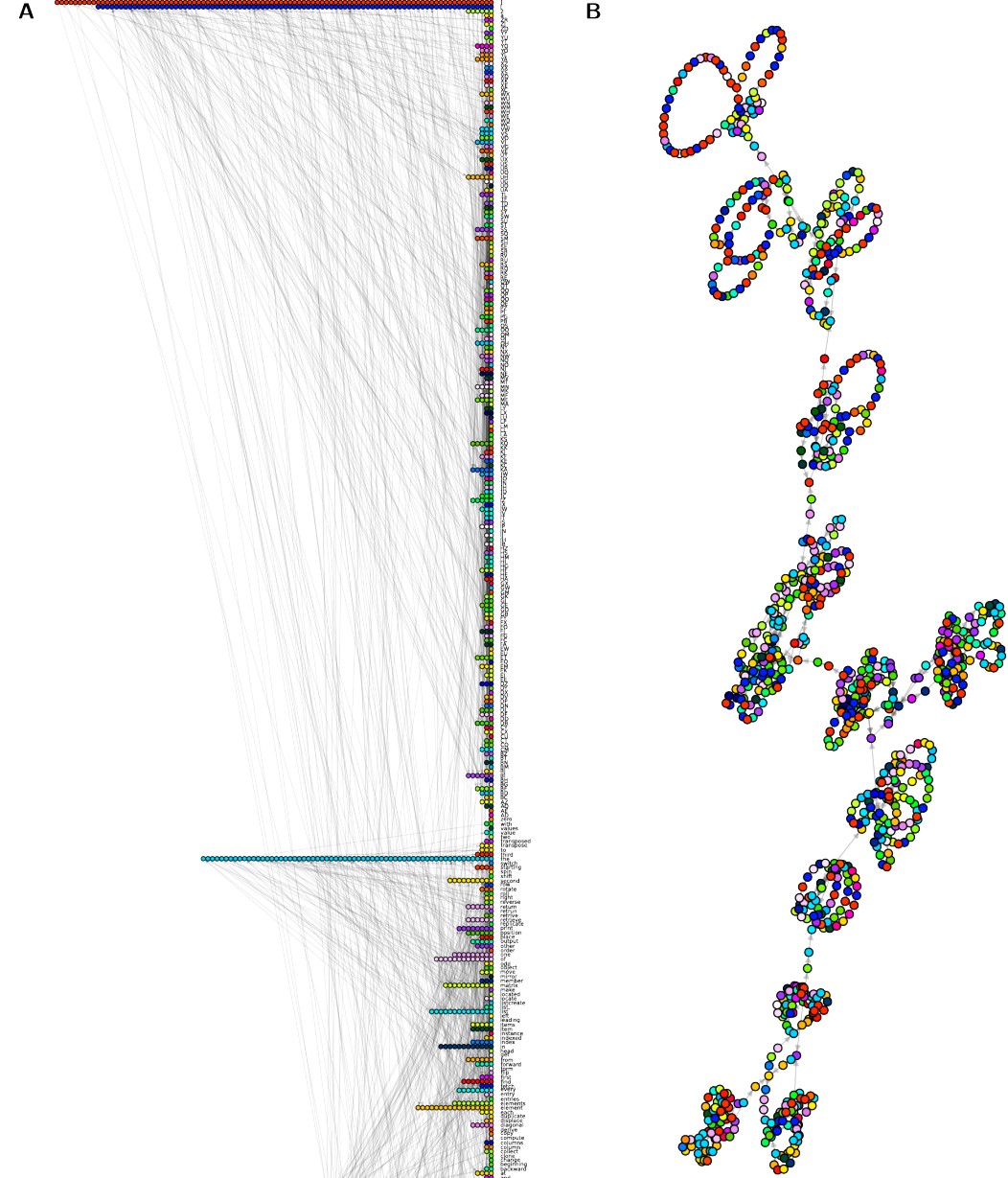

Figure 9: **A.** Transition graph of the learned CSCG model with overallocation ratio 3, visualized with stacked clones. **B.** The same transition graph visualized using the Kamada-Kawai algorithm [20] reveals 13 loosely connected clusters corresponding to the 13 algorithms used in the LIALT dataset.

## D.3  Results on the LIALT dataset

We present below the tables of results associated with Fig. 5. Our first Table 4 contains the in-context accuracies averaged on the entire test set.