# OpenReview forum: "Schema-learning and rebinding as mechanisms of in-context learning and emergence"
_NeurIPS.cc/2023/Conference — NeurIPS 2023 spotlight_

### Official Review · Reviewer_2Tdj · 2023-07-04

**Soundness:** 4 excellent
**Presentation:** 3 good
**Contribution:** 3 good
**Rating:** 7
**Confidence:** 3

**Summary:**

The authors attempt to analyze the reason for the success of In-context learning (ICL) for few-shot learning regimes. They apply ICL to clone-structured causal graphs (CSCGs) that can be used to interpret how ICL works in LLM. The CSCG is constructed as causal graph model where for the task of next token sequence prediction. The authors use CSCGs as interpertable language models to which they apply ICL to analyze key-properties of the learning mechanism. The authors go on to show how CSCG can perform similarly to ICL tasks designed for Transformer models and then show similar properties of CSCG to Transformer models (such as over-parameterization improves performance). The insights can help design new models.

**Strengths:**

1. The authors use CSCG an interpretable model to study a learning method (ICL) that is of high importance to the research community.
2. The experiments are thorough, and the authors show interesting capabilities of CSCG that they compare to a Transformer model.


**Weaknesses:**

1. It is a difficult paper to evaluate as the authors evaluate CSCG ability on standard ICL benchmarks such as 4.3 Dax test and the use of CSCG as a proxy to study ICL, e.g. 4.2 "Emergence". Expanding on my point, the authors choose to evaluate CSCG as a direct replacement to LLM (which does not appear to be either their premise or motivation of the work).
2. Related to 1.; the results and figures, lack conclusions. For example, I would expect to know how / what the experiment explains for how ICL works. An example where this is done successfully is Figure 5. `In-context accuracy (with standard errors) per task on the two LIALT test sets: for each task, overparametrization improves performance.`, but is not present in other figures or sections.
3. The jump on conclusions from a Transformer model to a CSCG is quite large. As such it appears the authors are under-delivering on their original premise, while the paper is of great interest without making the reference.

**Questions:**

Please see weaknesses.

---

> ### Author Rebuttal · Authors · 2023-08-09
>
> We appreciate the review for distilling the essential concepts we have tried to convey in the paper. Our goal here is indeed to elucidate a general framework for in-context learning behavior, leveraging the interpretability of CSCGs. A mechanistic understanding of ICL in transformers is still a work in progress (with several exciting works that we have referenced); we hope that this paper will spur further work in that direction. We shall emphasize this, and the current limitations of CSCGs as a direct alternative to LLMs, in the discussion section.
>
> > I would expect to know how / what the experiment explains for how ICL works. An example where this is done successfully is Figure 5, but is not present in other figures or sections.
>
> Thanks for the comment regarding clarity; we will improve the figure captions to make this more explicit.
>
> * In the GINC experiment, Section 4.1, we show that ICL is driven by (a) learning ​​a transition graph where the hidden variables are clustered by concepts (see Fig 3A) and (b) successful retrieval of the shared latent concept between the prompt and the model. In the first figure in our one-page PDF supplement to the global response, we illustrate the schema retrieval process, and the role played by increasing prompt length.
>
> * In the LIALT experiment, Section 4.2, we show that ICL requires — in addition to (a) and (b) — also (c) rebinding of novel tokens not seen during training to appropriate slots. Both Figs. 3B and 5[left] additionally show that model size drives retrieval and ICL capabilities. The second figure in our one-page supplement to the global response illustrates the schema retrieval and rebinding process, again leveraging the interpretability of CSCGs.

---

> > ### Comment · Reviewer_2Tdj · 2023-08-19
> >
> > I would like to thank the authors for their effort to perform the additional experiments for the figures. I have read their response and I will be keeping my score.

---

### Official Review · Reviewer_kc5K · 2023-07-06

**Soundness:** 4 excellent
**Presentation:** 4 excellent
**Contribution:** 4 excellent
**Rating:** 8
**Confidence:** 4

**Summary:**

This paper implements Cloned Structured Causal Graphs (CSCG), a model that was previously used in Neuroscience to explain Hippocampal cognitive maps, on language tasks that require in-context learning (ICL). They show the success of CSCG's across a variety of language benchmarks and come up with a theory on the mechanisms behind ICL and validate this theory using the results from ICL in CSCG's. The properties they claim are necessary for ICL include context-based separation, context-based merging for transitive generalization, and the presence of general abstract schema that can implement certain abstract content-independent operations that can also be rebound as necessary. Targeted experiments are conducted to illustrate each of these properties with a handful of relevant benchmarks, both old and new - GINC (old), LIALT (new), and PreCo (old).

**Strengths:**

* I have not seen CSCG's being used for the language domain, so I think that is a novel contribution inofitself.

* The properties that explain the mechanisms of ICL are very interesting and, although some I think can only be uniquely revealed through the architecture of CSCG's, seem like architecture-agnostic generalizable principles that add to the interpretability literature on ICL.

* Experiments are targeted, well-organized, and thorough. The contribution of the LIALT dataset also can be a valuable test-bed for ICL abilities.

**Weaknesses:**

The paper makes some (though I think reasonable) conjectures on how these properties are implemented in transformers. However, there isn't any empirical evidence to verify these claims. I don't think this is a strong weakness, though, because the empirical evidence from CSCG is a valid contribution by itself and this can be the topic of future work.

**Questions:**

What was the process of finding and identifying the schemas shown in Figure 2? Is there a principled way to detect abstract generalizable schemas learned during training?

**Limitations:**

I don't think the authors have included explicit sections discussing the limitations of using CSCG's in language modeling or on the broader societal impacts of this work. Would appreciate including this in a rebuttal, possibly with the extra page that is given to authors during the rebuttal phase.

---

> ### Author Rebuttal · Authors · 2023-08-09
>
> We are heartened by your emphasis on the strengths of the CSCG approach and the novelty of its application in this setting. We are similarly excited to leverage the interpretability of CSCGs to elucidate by analogy a general framework for in-context learning behavior. You might also find interesting the additional figures in our one-page PDF supplement to the global response, illustrating the process of schema retrieval and rebinding.
>
> > What was the process of finding and identifying the schemas shown in Figure 2?
>
> Fig. 2 aims at giving a high-level intuition of CSCG circuits to the reader: we selected simple sequence-to-sequence algorithms and manually designed the schemas. In Figs. 4 and 9, we show that we are able to extract similar schemas from a trained CSCG.
>
> > Is there a principled way to detect abstract generalizable schemas learned during training?
>
> Each schema corresponds to a cluster of latent states with a specific connectivity pattern. Performing community detection on the learned CSCG graph is one simple approach to discovering schemas.
>
> >  I don't think the authors have included explicit sections discussing the limitations of using CSCG's in language modeling or on the broader societal impacts of this work. Would appreciate including this in a rebuttal, possibly with the extra page that is given to authors during the rebuttal phase.
>
> We appreciate that you consider this of significant social impact to warrant discussion. In addition to elucidating the mechanics of ICL, we hope also that it serves as an exemplar in motivating the pursuit of interpretable models, especially as LLMs proliferate in application. Such interpretability will plausibly help tame biases and enforce guardrails for safety. We shall emphasize this, and the challenges of CSCGs for language modeling, in the discussion section.

---

> > ### Comment · Reviewer_kc5K · 2023-08-14
> >
> > Thank you for your response. I have read it and am happy to keep my rating.

---

### Official Review · Reviewer_ZCs7 · 2023-07-07

**Soundness:** 2 fair
**Presentation:** 2 fair
**Contribution:** 3 good
**Rating:** 6
**Confidence:** 1

**Summary:**

The paper aims to replicate the in-context learning (ICL) phenomenon in large language models (LLM) with clone-structured casual graphs (CSCG), which is roughly trying to learn hidden states in PODMP such that the transition matrix is invariant new environments, and only the emission matrix needs re-learning.

CSCG has the properties of context-separation, transitive generalization, schema-formation, and refining, which can be used to explain ICL behaviors. Results on three benchmarks show CSCG can match some LLM ICL behaviors.

**Strengths:**

- ICL is important and emerging, and its understanding is timely. The paper contributes to a novel view into ICL with some interesting constructions and experiments.

- The context-dependent latent representation and transitive generalization make sense.

- Experiments seem interesting and help replicate some ICL behaviors on toy datasets, e.g. accuracy of ICL depends on overparametrization.

**Weaknesses:**

- CSCG is unfamiliar to most readers and there could be some confusions that hinder understanding the paper (see questions). Importantly, **I do not understand how is the transition learned in the first space**, even before any "rebinding". And what is even the training data for each experiment??

- The setup of CSCG is still far from LLM ICL, and there should be acknowledgements of such limitations if the goal is to explain LLM ICL via CSCG. For example, learning and architecture, training task, test task...

I confess I do not understand CSCG fully. It might be a super smart and cool idea with a lot of potential, and I'm willing to raise my score these are addressed.

**Questions:**

- How is learning (of transition matrix) done? The paper seems to only talk about re-binding for learning the emission matrix...

- Why transition T is assumed to be fixed, and what kinds of train-test split do we assume this? (LLM might be hard to assume this?)

- Is the latent symbol space for z huge? Is "clone" just defined as #latent_symbols/#observed_symbols? Also, why not several latent tokens between observation tokens? Might make symbol space exponentially more efficient and more like language compositionality?

- Figure 3, is accuracy kind of low? does it really match Transformer models?

- Is this whole CSCG thing kinda like a latent automation? Do you think LLM is possible to recover such symbolic structures?

**Limitations:**

Did not see any limitation section in the paper, but there should be some (see weaknesses).

---

> ### Author Rebuttal · Authors · 2023-08-09
>
> Indeed, it is the combination of context-dependent latent representation and transitive generalization capability that drives the power of CSCGs. Since the thrust of this paper is on in-context learning behavior, page constraints limit us from elaborating on CSCG details. We are happy to add to the appendix a section elaborating on these details, if that might be helpful. Towards the end of this response, we try to clarify the aspects you have asked about.
>
> Our emphasis in this paper has been on leveraging the interpretability of the CSCG model for elucidating in-context learning behavior. While we do not directly interpret ICL behavior in LLM models, we reason by analogy, and hope that this paper will spur more work in that direction. A mechanistic understanding of ICL in transformers is still a work in progress (with several exciting works that we have referenced). We hope also that the interpretability of CSCGs will motivate such an emphasis in LLMs (and sequence models more generally) as they proliferate in application. We will emphasize these aspects in the discussion section.
>
>
> > Also, why not several latent tokens between observation tokens? Might make symbol space exponentially more efficient and more like language compositionality?
>
> You seem to be suggesting having a CSCG with factorized latent space: this is a relevant modification to the model. We agree that such a structure can allow better compositionality while enabling scalability. We hope to explore such modifications in our future research.
>
>
> > Figure 3, is accuracy kind of low? Does it really match Transformer models?
>
> On the GINC dataset, CSCG achieves an in-context accuracy above 90% for context-length of 5 and above 95% for context lengths of 8 or 10.
> These results are higher than the ones reported for Transformers in [2]-which introduced the GINC dataset–and comparable to the best LSTMs reported by the authors. We acknowledge that further tuning might boost transformers’ performance; we are only conveying that these high accuracies illustrate that CSCGs are good at context-specific retrieval (which is essentially what the GINC dataset is probing).
>
>
> > Is this whole CSCG thing kinda like a latent automation? Do you think LLM is possible to recover such symbolic structures?
>
> You are right, one could think of the action-conditioned transition matrix as a stochastic generalization of a finite deterministic automaton in the latent space.
> Some recent work [4] has demonstrated how LLMs might be able to learn such automata.  The advantage of CSCGs is how easily interpretable the learned schemas are, as graphs. Refer the second figure in our supplemental PDF to the global response, for an example.
>
>
> CSCG details:
> * Train & test data: For the GINC dataset, we use the same training and in-context test set as [2], which is available on GitHub [3]. For the LIALT dataset, we describe the structure of the train and test set (with examples) in fig. 4A, and lines 174-184 and lines 185-193 of the text. One can think of our training sets as analogous to the pretraining datasets used in LLMs, and of our in-context test set as the prompts used to test LLMs for in-context learning.
> * Learning of the transition matrix: Learning in CSCGs uses the Expectation-Maximization algorithm to maximize the log-likelihood of the CSCG on a sequence of symbols. It is therefore similar to HMMs training, with additional computational benefits due to the clone structure. Note that during CSCG training, the emission matrix (which specifies the latent spaces associated with an observation) is fixed, and only the transition matrix is learned. The original CSCG paper [1] has more details.
> * Freezing of the transition matrix: The transition matrix T is only learned during the “training” phase, which we describe in line 155 (for the GINC dataset) and in line 194 (for the LIALT dataset). We demonstrate that ICL behavior in CSCGs,  which emulate  “algorithms” on novel tokens, can be implemented by (a) freezing the learned transition matrix (the schemas, corresponding to algorithms) at test time, and (b) only modifying the emission matrix through the rebinding algorithm.
> * Latent space: “Clones” correspond to different latent states which emit the same observation. For a given dataset, the latent space that needs to be modeled is at least as big as the number N of distinct contexts necessary to correctly predict the next token.  In line 196, we introduce an “overallocation ratio” to parameterize the CSCG capacity proportionally to N. In practice, our latent space (including the overallocation) contains a few thousand states.
>
>
> References:
>
> [1] Dileep George et al. “Clone-structured graph representations enable flexible learning and vicarious evaluation of cognitive maps”. In: Nature communications 12.1 (2021), p. 2392.
>
> [2] Sang Michael Xie et al. “An explanation of in-context learning as implicit bayesian inference”. In: arXiv preprint arXiv:2111.02080 (2021)
>
> [3] https://github.com/p-lambda/incontext-learning/blob/main/data/GINC_trans0.1_start10.0_nsymbols50_nvalues10_nslots10_vic0.9_nhmms10
>
> [4] Bingbin Liu et al. “Transformers learn shortcuts to automata”. In: arXiv preprint arXiv:2210.10749 (2022)

---

> > ### Comment · Reviewer_ZCs7 · 2023-08-22
> > **Thanks**
> >
> > I've increased my score from 5 to 6, but I hope authors work on clarity during revision to make the interesting topic more accessible and understandable!

---

### Official Review · Reviewer_PPgs · 2023-07-09

**Soundness:** 2 fair
**Presentation:** 3 good
**Contribution:** 3 good
**Rating:** 7
**Confidence:** 3

**Summary:**

The authors propose the clone-structured causal graph (CSCG) with rebinding as a model of in-context learning in language models. The authors conduct several experiments showing that CSCGs can learn latent graphs corresponding to meaningful concepts seen in the training data, while also generalizing to instantiations of those concepts containing novel tokens seen at test time.

**Strengths:**

The presentation of the CSCG as a model of in-context learning is new to me and very interesting, addressing a hot topic in LLMs that is relevant to a large portion of the community. The experiments show that CSCGs can be learned in a way that leads to meaningful concepts, as well as the fact that overparameterization is useful for CSCGs as with typical LLMs. Finally, the experiments show that CSCGs can perform fast binding of novel tokens to previously-seen structures in a "dax test", which is an important capability the authors argue is not adequately explained by existing frameworks.

**Weaknesses:**

Key claims about interpretability are not substantiated. In particular, the abstract and intro clearly emphasize the importance of interpretability in models of language/ICL, but the experiments do not meaningfully explore how CSCGs are more interpretable than existing LLMs.

Some experiments are somewhat superficial or small scale, particularly 4.3, where only a few qualitative examples are shared, without any quantitative analysis showing that the results are representative. The largest datasets are orders of magnitude smaller than the datasets used by real LLMs. This shortcoming isn't absolutely critical, but of course the results would be much more convincing if ICL phenomena could be shown at something closer to real-world scale.

**Questions:**

How does the size of the latent graph for e.g. list reversal scale with list size?

What is the computational cost of fitting a CSCG with EM? PreCo is many orders of magnitude smaller than typical pre-training corpora. Is CSCG plausible even for very large corpora?

Are the results in Figure 6 cherry picked? How often is rebinding + MAP inference successful?

Is there a typo in the emission matrix/clone structure definition? The number of columns of the emission matrix is the size of the observation space when it is defined, but it is the size of the latent space when the clone structure condition is defined later in 2.1.

**Limitations:**

The authors should add some discussion about the limitations of CSCGs for modeling the behaviors of real-world LLMs, particularly whatever computational challenges may exist in scaling CSCG-like models to billions or trillions of tokens.

---

> ### Author Rebuttal · Authors · 2023-08-09
>
> We are glad that you consider novel and interesting our application of CSCGs towards a framework for understanding ICL. We will elaborate on the challenges of scaling CSCGs to large datasets in the discussion section, and some potential directions for progress.
>
> To recap how we leverage CSCG interpretability in the paper: in Fig. 3A, we extract the CSCG transition graph and observe five clusters – corresponding to each of the concepts in the GINC dataset. Similarly, in Fig. 4B, we extract interpretable circuits from a CSCG trained on the LIALT dataset: these circuits explicitly represent sequences of different lengths (relating also to the question of the latent graph size).
>
> We also present two additional figures in the one page PDF supplement to the global response (which we will include in the final paper), illustrating:
> * How increasing prompt length contributes to schema retrieval
> * The step-by-step functioning of the retrieval and rebinding process, on LIALT prompts
>
> Finally, please note that some previous work on CSCGs [1] has also exploited their interpretable structure to match with cognitive maps in the hippocampus.
>
> Cumulatively, we hope this buttresses our claims regarding the interpretability of CSCGs.
>
>
> > How does the size of the latent graph for e.g. list reversal scale with list size?
>
> The size of the latent graph for the reversal algorithm, as measured by the number of nodes, scales (a) as O(N) when the list size is fixed to N (b) in the worst case, as O(N^2) when the list sizes can vary between 1 and N. We illustrate this scaling (a) conceptually in Figs. 2E and 2F and (b) empirically by extracting the latent graph from a trained CSCG in Fig. 4B: the unrolled view illustrates how lists of different sizes are represented in the latent graph.
>
>
> > What is the computational cost of fitting a CSCG with EM? [...] Is CSCG plausible even for very large corpora?
>
> For a CSCG with M clones per symbol trained on a sequence of length N, each EM iteration has a computational cost in O(N M^2). Note that the size of the latent space is H = M*E, where E is the total number of distinct symbols. It is possible to exploit the sparsity structure of the problem (as well as other elements, such as factorizing the transition matrix, parallelizing the computation of the EM steps, etc)  to scale the training of CSCGs to large language datasets.
>
>
> > Are the results in Figure 6 cherry picked? How often is rebinding + MAP inference successful?
>
> We selected the examples in Figure 6 so that, when filling in the words, the context makes it clear what the missing word is. The same is necessary for humans; it is hard to rebind correctly with an ambiguous sentence such as “I went to the dax this morning”. When there is no ambiguity about the missing word, we found MAP inference + rebinding to be successful.
>
>
> > Is there a typo in the emission matrix/clone structure definition? The number of columns of the emission matrix is the size of the observation space when it is defined, but it is the size of the latent space when the clone structure condition is defined later in 2.1.
>
> Thank you for pointing out a typo on line 64: we have inverted the rows and columns of the emission matrix. We will fix it.

---

> > ### Comment · Reviewer_PPgs · 2023-08-16
> >
> > I appreciate the authors' response. If the proposed revisions are included in the final paper, I am happy to keep my score.

---

### Author Rebuttal · Authors · 2023-08-09

We would like to thank the reviewers for their time and thoughtful comments. As the reviews have identified, we have used CSCGs as a sequence model and leveraged their interpretability to deconstruct in-context learning (ICL) behavior into a combination of schema-learning (at training time) and schema-retrieval + rebinding (at prompting time). We hope our work demystifies by analogy the surprising ICL behavior observed in LLMs, showcases avenues for further research on ICL capabilities, and provides impetus for interpretable methods.

In our one page supplement, we present two additional figures that illustrate the mechanics of the schema retrieval and rebinding process, and also highlight the interpretability of CSCGs. We shall make use of the additional page for the camera-ready version of the paper to incorporate these.

---

### Comment · Area_Chair_QR92 · 2023-08-10
**Author-Reviewer Discussion phase (Aug 10-16)**

Today begins the Author-Reviewer Discussion phase, which lasts 1 week (**Aug 10-16**).

I ask the reviewers to please **carefully read all other reviews and the author responses
and (if appropriate) respond to author responses promptly.**   If you've read the author response, please take the time to leave a comment, even if you have nothing to add.

I also encourage both authors and reviewers to monitor OpenReview for further comments in order to enable as much back-and-forth as possible during this short period.

---

### Decision · Program_Chairs · 2023-09-21

**Decision:**

Accept (spotlight)

**Comment:**

This work shows how in-context learning can emerge in clone-structured causal graphs (CSCGs).  Reviewers observed that the conjectured connections with transformers (e.g. Discussion, Section 6) were not empirically supported, and that the claims of greater interpretability were not supported.  Despite these issues, reviewers rated the paper highly, and universally favored acceptance.

I recommend acceptance, and I also believe that Section 6 adds to the paper, as it highlights an important advantage this work has over several previously proffered explanations of in-context learning.  However, this section can be improved in clarity, and I believe the authors should clarify how CSCGs relate to induction heads, which they note are another explanation for in-context learning, but do not discuss in any depth.  Furthermore, the claims around interpretability should be tempered, and practical limitations of the work should be noted, in line with the comments of Reviewer PPgs.